# Finding Waldo: The Evolving Paradigm of Circulating Tumor DNA (ctDNA)—Guided Minimal Residual Disease (MRD) Assessment in Colorectal Cancer (CRC)

**DOI:** 10.3390/cancers14133078

**Published:** 2022-06-23

**Authors:** Sakti Chakrabarti, Anup Kumar Kasi, Aparna R. Parikh, Amit Mahipal

**Affiliations:** 1Department of Hematology-Oncology, Medical College of Wisconsin, 8701 W Watertown Plank Rd, Milwaukee, WI 53226, USA; 2Division of Medical Oncology, University of Kansas, Kansas City, KS 66160, USA; akasi@kumc.edu; 3Harvard Medical School, Massachusetts General Hospital Cancer Center, Boston, MA 02114, USA; aparna.parikh@mgh.harvard.edu; 4Mayo College of Medicine, Rochester, MN 54656, USA; mahipal.amit@mayo.edu

**Keywords:** circulating tumor DNA, colon cancer, colorectal cancer, minimal residual disease, adjuvant chemotherapy

## Abstract

**Simple Summary:**

After the surgical removal of colorectal cancer (CRC), residual cancer cells undetectable by standard blood tests and imaging studies are responsible for cancer recurrence. Currently, chemotherapy is often administered after surgery to eradicate residual cancer cells, a decision guided by clinical and pathologic criteria, which are imprecise. Circulating tumor DNA (ctDNA) consists of DNA fragments in the bloodstream derived from cancer cells, and the presence of ctDNA likely indicates the presence of residual cancer cells. The current article discusses how ctDNA technology can help guide treatment in patients with CRC after curative surgery.

**Abstract:**

Circulating tumor DNA (ctDNA), the tumor-derived cell-free DNA fragments in the bloodstream carrying tumor-specific genetic and epigenetic alterations, represents an emerging novel tool for minimal residual disease (MRD) assessment in patients with resected colorectal cancer (CRC). For many decades, precise risk-stratification following curative-intent colorectal surgery has remained an enduring challenge. The current risk stratification strategy relies on clinicopathologic characteristics of the tumors that lacks precision and results in over-and undertreatment in a significant proportion of patients. Consequently, a biomarker that can reliably identify patients harboring MRD would be of critical importance in refining patient selection for adjuvant therapy. Several prospective cohort studies have provided compelling data suggesting that ctDNA could be a robust biomarker for MRD that outperforms all existing clinicopathologic criteria. Numerous clinical trials are currently underway to validate the ctDNA-guided MRD assessment and adjuvant treatment strategies. Once validated, the ctDNA technology will likely transform the adjuvant therapy paradigm of colorectal cancer, supporting ctDNA-guided treatment escalation and de-escalation. The current article presents a comprehensive overview of the published studies supporting the utility of ctDNA for MRD assessment in patients with CRC. We also discuss ongoing ctDNA-guided adjuvant clinical trials that will likely shape future adjuvant therapy strategies for patients with CRC.

## 1. Introduction

Colorectal cancer (CRC) remains the third most frequently diagnosed cancer in the United States (US) [1]. Approximately 150,000 individuals are diagnosed with CRC each year in the US [1], and by 2030, the global burden of CRC is expected to increase by 60% [2]. The incidence pattern of CRC has shown a divergent trend in the last two decades, with rising incidence in the population younger than 50 years that poses unique challenges [3]. According to a recent study, by 2040, CRC is expected to be the second most frequently diagnosed cancer and the most common cause of cancer-related death in adults aged 20 to 49 years [4]. It is noteworthy that approximately 80% of CRC patients present with an early-stage disease amenable to curative-intent therapy [5], and 25 to 40% of these patients endure cancer recurrence despite definitive treatment [6]. Consequently, advancement in the treatment of early-stage CRC will impact the overall mortality from CRC. 

Minimal residual disease (MRD) refers to the neoplastic cells below detectable levels with conventional methods that persist in the body after the completion of definitive therapy (surgery with or without adjuvant chemotherapy (ACT)) [7]. An overwhelming amount of research suggests that MRD is responsible for cancer recurrence [8]. For many decades, clinicopathologic risk factors have served as a surrogate for MRD and dictated the patient selection strategy for ACT in early-stage CRC patients [9]. However, accumulating data indicate that risk stratification based on the clinicopathologic criteria is imprecise in identifying patients with MRD, resulting in overtreatment and undertreatment of a significant number of patients [10,11,12]. To elaborate further, it is well known that approximately 50% of the stage III patients and 74% of low-risk stage III patients are cured by surgery alone [10,13]. However, current guidelines [14,15] recommend ACT for all such patients, resulting in unnecessary chemotherapy administration in a large proportion of patients, causing a myriad of short- and long-term toxicities. Conversely, 5% of stage I CRC and 10–25% of stage II patients recur; however, for stage I and average-risk stage II patients, there is no available risk-stratification tool to identify patients who are destined to recur after definitive surgery [16]. Furthermore, adjuvant trial results suggest that the survival benefit with oxaliplatin-based ACT is modest, adding approximately 17% absolute 5-year disease-free survival (DFS) benefit over surgery alone in stage III patients (i.e., oxaliplatin-based ACT increases 5-year DFS from around 50% with surgery alone to 67% with surgery plus ACT [10,13,17,18,19]). It is also important to emphasize that the benefit of ACT after modern surgery may be less than quoted above, as the improvements in surgical techniques possibly cure more patients than before [20,21,22]. For example, excision of a higher number of lymph nodes, a strategy vigorously promoted by the major guidelines and pursued by modern surgeons, correlates with improved survival in stage II and III colon cancer patients [23]. These data highlight that the benefits of ACT in patients with early-stage CRC undergoing modern surgery are likely limited to the highest-risk patients [13], underscoring the urgent need for a biomarker that can precisely detect MRD and refine the patient selection strategy for adjuvant therapy administration.

The lower absolute risk of cancer recurrence with modern surgery and the risk of long-term toxicities, especially neurotoxicity, associated with oxaliplatin use [18,19] prompted the IDEA [24], an international collaboration examining the adjuvant therapy de-escalation strategy that resulted in a 3-month duration of ACT for most subgroups of resected colon cancer patients [25]. A precise and reproducible MRD detection tool would further add to the ongoing treatment de-escalation effort, an unmet need that has impeded the progress of adjuvant therapy research for many decades. 

Circulating tumor DNA (ctDNA), the tumor-derived single- or double-stranded DNA fragments detectable in the plasma, has emerged as a promising biomarker for MRD assessment in patients with CRC who have undergone curative-intent surgery or completed adjuvant therapy [26]. Several single-arm prospective studies have reported a high degree of correlation between the presence of ctDNA after the completion of definitive treatment and cancer relapse [27,28,29,30,31,32,33,34]. These studies suggest that ctDNA-based MRD assessment outperforms all existing clinicopathologic criteria-based risk-stratification strategies. The current article provides an overview of the clinical utility of ctDNA-based MRD assessment that can influence adjuvant therapy decisions. We also discuss the ongoing clinical trials investigating various ctDNA-guided adjuvant treatment strategies.

## 2. Targeting MRD: Rationale and Evidence Supporting the Strategy

A plethora of research suggests that micrometastatic disease, or MRD, is responsible for cancer recurrence in patients who have completed curative-intent treatment [35]. Adjuvant therapy aims to eliminate MRD and achieve a cure. Several conceptual arguments support the strategy of targeting MRD before clinically overt cancer relapse [7]. First, complete eradication of MRD to achieve a cure requires eradicating all neoplastic cells, including the ‘cancer stem cells’, which are less sensitive to anticancer therapies than differentiated cancer cells [36]. It might be easier to eradicate cancer stem cells in the MRD stage when cancer stem cells are less numerous than in the clinically overt relapse stage, as cancer stem cells constitute a small fraction of the total neoplastic cell burden. Second, as clonal complexity and drug resistance correlate with the total neoplastic cell burden, the likelihood of having numerous drug-resistant clones at the MRD stage is lower [37]. Third, the small number of cancer cells in the MRD stage is unlikely to induce significant microenvironment remodeling and build robust chemoprotective niches, and therefore, anticancer agents might be more effective in the MRD stage [38]. Finally, patients are likely to have superior performance status, and be more tolerant to anticancer therapy during the asymptomatic MRD stage than during the overt cancer relapse.

It is well-established that adjuvant therapy intended to eliminate MRD increases the probability of cure in many solid tumors. The ‘evidence for cure’ supporting the strategy of treating patients before clinically overt cancer relapse (i.e., adjuvant therapy) abounds in the literature, both for colon cancer [18,39] and other solid tumors [40]. A logical extension of this strategy is tailoring treatment based on the MRD burden, a well-established practice in the treatment of childhood acute lymphoblastic leukemia (ALL) that improves survival [41]. Targeted therapies directed at molecular MRD in chronic myeloid leukemia and ALL harboring BCR–ABL fusion kinase [42] have been an integral part of routine care for many decades. In the solid tumor space, adjuvant therapy with osimertinib targeting activating mutations in epidermal growth factor receptor (EGFR) in patients with completely resected, EGFR-mutated non-small cell lung cancer [43] is another example of the successful targeting of molecular MRD that improves survival. Therefore, a tool that can accurately detect and measure the MRD burden to enable adjuvant therapy administration tailored to the MRD burden will be critical in transforming the adjuvant therapy paradigm of CRC.

## 3. Biology of Cell-Free DNA

Cell-free DNA (cfDNA), the extracellular DNA fragments detectable in various body fluids, including plasma, is mainly derived from the hematopoietic system due to cell death and normal cellular turnover [44]. The processes involved in cfDNA shedding from cells are necrosis, apoptosis, active secretion, pyroptosis, autophagy, and NETosis [45]. ctDNA is the tumor-derived cell-free DNA fragments in the plasma, composed of short fragments ranging from 130 to 150 base pairs released from the cancer cells via necrosis, apoptosis, and active secretion [46]. It is well recognized that cancer patients have higher plasma levels of cfDNA than the healthy population [37,38]. The fraction of ctDNA in plasma varies widely in cancer patients, ranging from less than 0.1% to greater than 10%, depending on various factors, including tumor burden, shedding characteristics of the tumor, and anatomic site of the tumors [47,48,49]. Specifically, tumors confined to the lungs or peritoneum are associated with significantly lower levels of ctDNA [47]. Furthermore, several non-malignant conditions, including acute trauma, surgical procedures, ischemia, infection, or inflammation, can increase the cfDNA level and can potentially interfere with ctDNA assays [50].

Observational studies indicate that the half-life of ctDNA in circulation is approximately 2 h [49], suggesting that the ctDNA level provides a ‘real-time’ snapshot of the tumor dynamics. cfDNA is cleared from the circulation via nuclease activity, degradation by macrophages in the liver and spleen, and renal excretion into the urine [46]. Of note, DNA fragments derived from malignant cells have a shorter fragment length than DNA fragments derived from dying normal cells, which enables the improved detection of ctDNA by selecting fragments between 90 bp and 150 bp [51]. Plasma samples are preferable to serum samples for ctDNA testing, as the serum is typically contaminated with DNA derived from lysed leukocytes during the clotting process, which interferes with the assay [52].

## 4. ctDNA Assays

An exquisitely sensitive assay is needed to detect DNA fragments derived from the residual neoplastic cells after curative-intent surgery, as ctDNA in the post-operative setting typically constitutes <0.01% of cfDNA [53]. In resected CRC patients, Reinert et al. reported a median of three tumor molecules per milliliter of plasma in ctDNA-positive samples [29]. Furthermore, trauma induced by the surgery increases the cfDNA level in the plasma for up to 4 weeks, making ctDNA detection challenging [53]. Consequently, the analytical sensitivity of the ctDNA test and the sample timing significantly influence the ctDNA detection rate in the setting of MRD. ctDNA assays utilized in the major prospective studies in patients with resected CRC belong to two broad categories: (a) tumor-agnostic assays and (b) tumor-informed assays. Table 1 summarizes the salient features of the most widely studied ctDNA assays.

Tumor-agnostic assays are broad panel-based sequencing assays performed without prior knowledge of the patient’s tumor mutational profile and designed to look for genomic alterations and aberrant DNA methylation patterns known to occur in a given tumor type (e.g., Guardant REVEAL) [28]. These assays include aberrant methylation patterns to the somatic mutation panel to improve sensitivity, as aberrant DNA methylation is often an early step in the carcinogenesis of CRC [54]. Tumor-agnostic assays have several advantages that include fast turnaround time, logistical simplicity, ability to perform the test if the primary tumor tissue is not available, and the potential of detecting MRD even after clonal evolution of the micrometastatic tumor cells.

Conversely, tumor-informed assays require prior knowledge of the tumor genomic profile of the index patient, generally acquired by whole-exome sequencing or targeted sequencing of the primary tumor (e.g., Signatera^TM^, SafeSeqS) [29,55]. These assays are personalized and designed for each patient to detect patient-specific genomic alterations via the targeted sequencing of the plasma DNA. Tumor-informed assays can also be designed on the droplet digital PCR (ddPCR) platform [30], in which a droplet generator partitions plasma samples into numerous discrete droplets, ensuring that each droplet contains no more than one fragment of the template DNA, followed by simultaneous analysis for target sequences through an endpoint PCR. Tumor-informed assays have several advantages, including a high level of analytical sensitivity down to a variant allele frequency of 0.01% [29] and a low probability of false-positive results secondary to clonal hematopoiesis of indeterminate potential (CHIP) [56]. However, tumor-informed assays require a longer turnaround time and incur additional costs for tumor sequencing. Furthermore, tumor sequencing may not capture all MRD relevant alterations due to intratumoral heterogeneity [46], and may not detect emerging mutations arising from treatment-related selection pressure [57].

**Table 1 cancers-14-03078-t001:** ctDNA assay platforms utilized to assess minimal residual disease (MRD) in major prospective studies in CRC.

ctDNA Assay	Tumor-Informed	Assay Description	Target Alterations in Plasma DNA	Turnaround Time	Comments
Guardant REVEAL [28]	No	Plasma-only NGS-based test that integrates somatic alterations and epigenomic cancer signatures.	Somatic and epigenetic aberrations	2 weeks	Integrating epigenomic signatures increased sensitivity by 25–36% versus genomic alterations alone. **Fastest turnaround time as tumor sequencing is not required.**
Safe-seqS [55]	Yes	Tumor sequencing followed by deep sequencing of plasma DNA with unique molecular barcoding to detect tumor-specific mutations.	One tumor-specific mutation in each patient	2 weeks *	ctDNA result is classified as detectable (ctDNA-positive) or undetectable
Signatera [29]	Yes	A personalized, tumor-informed, multiplex PCR-based NGS assay. Sixteen patient-specific, somatic SNVs are selected for each patient based on the whole-exome sequencing of the tumor for interrogation in the cfDNA. Plasma samples with at least two tumor-specific SNVs are defined as ctDNA-positive.	16 somatic variants	7–10 days *	Limit of detection 0.01% variant allele frequency. Provides ctDNA level expressed as mean tumor molecules (MTM)/mL of plasma
ddPCR [58]	Yes	Targeted sequencing of the primary tumor for a predefined panel of 29 genes followed by an interrogation of plasma cfDNA by ddPCR to search for the tumor-specific mutations (1–2 mutations).	1 to 2 alterations selected by tumor sequencing	2–5 days *	Tracking at least two variants in plasma increased the ability to identify MRD to 87.5%.

Abbreviations: ctDNA, circulating tumor DNA; NGS, next-generation sequencing; PCR, polymerase chain reaction; SNV, single nucleotide variant; cfDNA, cell-free DNA; ddPCR, droplet digital PCR. * Reflects the time required to generate the result after the assay has been designed. The first report may take up to four weeks.

## 5. ctDNA-Guided MRD Assessment: Studies in Colorectal Cancer

Over a decade ago, a study reported by Diehl et al. first suggested that ctDNA could be a surrogate marker for MRD in resected CRC [49]. This study obtained serial blood samples from 18 patients with CRC and liver metastasis undergoing resection. All but one of the 12 patients with detectable ctDNA post-operatively had cancer relapse. Conversely, none of the four patients with undetectable ctDNA in the post-operative plasma samples had a recurrence. This study inspired a series of subsequent prospective studies that provided critical evidence supporting the value of ctDNA-guided MRD assessment in patients with resected CRC (summarized in Table 2).

Several seminal studies published by Tie and colleagues provided a wealth of data supporting the clinical utility of ctDNA for MRD assessment in CRC patients [31,32,33,55,60]. In their studies, Tie and colleagues used the SafeSeqS assay, a personalized, tumor-informed ctDNA assay designed to look for a panel of tumor-specific genomic alterations in the plasma DNA by means of targeted sequencing [55]. The first study in this series was a single-arm observational study with stage II colon cancer patients (*n* = 230) in which serial blood samples were obtained for ctDNA testing starting 4–10 weeks after the surgery and every three months thereafter for two years [55]. In this study, among the 178 patients who did not receive ACT, ctDNA detection in the immediate post-operative period (i.e., 4 to 10 weeks after surgery) predicted cancer recurrence in all patients (estimated 3-year relapse-free survival (RFS) of 0%), while patients with undetectable ctDNA in the post-operative period had a 3-year RFS of 90% (HR, 18; *p* < 0.001). In patients receiving ACT, the presence of ctDNA following completion of ACT also predicted an inferior RFS (HR, 11; *p* = 0.001). In addition, this study reported a much higher sensitivity and specificity of ctDNA than carcinoembryonic antigen (CEA) in predicting radiological cancer recurrence, with ctDNA being positive in 85% of patients at the time of radiological recurrence vs. CEA being elevated in 41% of patients (*p* = 0.003). Tie and colleagues subsequently reported similar findings in patients with stage III colon cancer [32], resected metastatic CRC [60], and locally advanced rectal cancer undergoing multimodality treatment [33]. A subsequent pooled analysis of studies with nonmetastatic CRC [31] (*n* = 485: 230 stage II colon cancer, 96 stage III colon cancer, and 159 locally advanced rectal cancer) reported inferior 5-year RFS (38.6% vs. 85.5%; *p* < 0.001) and OS (64.6% vs. 89.4%; *p* < 0.001) in patients with detectable ctDNA after the completion of definitive therapy, confirming consistent long-term prognostic impact of ctDNA. Furthermore, studies by Tie and colleagues demonstrated that ctDNA outperformed all existing clinicopathologic prognostic factors.

Two single-arm prospective studies with non-metastatic CRC patients utilizing a different personalized, tumor-informed ctDNA assay platform, Signatera^TM^, provided more data boosting the idea of ctDNA-guided MRD assessment [27,29]. In the study reported by Reinert et al. with 130 stage I to III CRC patients, positive ctDNA at post-operative day 30 was associated with a seven times higher risk of cancer recurrence compared to ctDNA-negative patients (HR, 7.2; *p* < 0.001) [29]. Similarly, positive ctDNA immediately after ACT and during surveillance was associated with 17 times (HR, 17.5; 95% CI, 5.4–56.5; *p* < 0.001) and 40 times (HR, 43.5; 95% CI, 9.8–193.5 *p* < 0.001) higher risk of cancer relapse, respectively. On multivariable analyses, ctDNA was independently associated with the risk of cancer recurrence after adjusting for known clinicopathologic risk factors. Furthermore, this study reported recurrence rates as low as 3%, with a negative predictive value of 97%, if serial ctDNA tests were negative for three years. An extension of this study with stage III CRC patients reported similar findings [27]. Congruently, studies utilizing tumor-informed ddPCR-based assays [30,59], studies in locally advanced rectal cancer [33,59], and studies in metastatic CRC undergoing curative-intent treatment [60,61] reported similar findings reiterating a robust prognostic impact of positive ctDNA after the surgery and completion of adjuvant/definitive therapy.

Parikh and colleagues published pioneering work with a tumor-agnostic plasma-only ctDNA assay platform (REVEAL by Guardant Inc., Redwood City, CA, USA) that integrates genomic and epigenomic tumor signatures [28]. In a single-arm observational study with the REVEAL platform, 84 of 103 patients had evaluable results after completion of definitive therapy. All 15 patients with detectable ctDNA one month after definitive treatment and more than one year of follow-up had cancer recurrence (positive predictive value, 100%). Furthermore, this study reported improved sensitivity with longitudinal sampling and integration of epigenomic signatures into the assay. Of note, the turnaround time of this assay is impressively short, at approximately two weeks.

Finally, recently presented interim results of the GALAXY study utilizing the tumor-informed Signatera^TM^ assay platform propelled the ctDNA-guided MRD assessment paradigm to a new level [62]. GALAXY is the observational arm of the ongoing CIRCULATE-Japan platform study designed to evaluate the clinical utility of ctDNA in determining MRD status in patients with resected clinical stage II to IV CRC. In this study, blood samples were collected before surgery, one month after surgery, and every three months thereafter for two years. The results have been reported after a median follow-up of 11.4 months. Congruent with the previous studies discussed above, the risk of cancer relapse in patients with resected stage II and III CRC had a high degree of correlation with ctDNA positivity at one month after surgery (HR, 13.3; *p* < 0.001). The GALAXY study also reported valuable information regarding the impact of ACT on ctDNA clearance. ctDNA clearance was observed in 68% of patients who were ctDNA-positive at one month post-surgery and received oxaliplatin-based ACT. Patients who cleared their ctDNA with ACT had similar survival outcomes as those who were ctDNA-negative post-surgery (HR, 0.8; *p* = 0.60), although the follow-up period was relatively short (11.4 months). Conversely, patients who did not clear ctDNA with ACT had 15.8 times increased risk of cancer recurrence than those who cleared their ctDNA (*p* ≤ 0.001). Patients with negative ctDNA four weeks after surgery had excellent survival outcomes irrespective of ACT administration, with a DFS of approximately 95% at 12 months. These data provide a preliminary glimpse of the potential of ctDNA technology in guiding patient selection for adjuvant therapy. However, it is important to highlight that the median follow-up of this study is relatively short (11.4 months) at this point, and longer-term data are necessary before the study findings can be incorporated into practice guidelines.

Recently published DYNAMIC trial results utilizing the SafeSeqS assay further support the ctDNA-guided adjuvant therapy paradigm [63]. DYNAMIC was a prospective randomized phase II study to assess whether ctDNA-guided adjuvant therapy in resected stage II colon cancer patients could reduce ACT use without increasing recurrence risk. In this study, resected stage II colon cancer patients were randomized in a 2:1 ratio to ctDNA-guided ACT versus standard clinicopathologic factor-guided ACT. The primary study endpoint was RFS at two years, and a key secondary endpoint was ACT use. Of the 455 randomized patients, 302 were assigned to ctDNA-guided treatment and 153 to standard management. After a median follow-up of 37 months, a much lower proportion of patients in the ctDNA-guided group received ACT compared to the standard management group (15% vs. 28%; relative risk, 1.82; 95% CI, 1.25 to 2.65) without any detriment to 2-year RFS in the ctDNA-guided group (93.5% and 92.4%, respectively; 95% CI, −4.1 to 6.2 [noninferiority margin, −8.5 percentage points]). Furthermore, among the ctDNA-positive patients, 3-year RFS was 92.6% in patients receiving oxaliplatin-based ACT and 76.0% in patients receiving single-agent fluoropyrimidine. This trial result will pave the way for the ctDNA-guided adjuvant therapy approach in the routine clinical practice guidelines. Figure 1 summarizes the risk of cancer recurrence with positive ctDNA in post-operative and post-ACT settings reported in major prospective studies.

It is important to emphasize that ctDNA overwhelmingly outperforms CEA in predicting cancer relapse in patients with resected CRC. A large retrospective study reported a 40% false-positivity rate of modestly elevated CEA (≥5.1 ng/mL but ≤10.1 ng/mL) in resected CRC patients [64]. Several prospective studies described above evaluated ctDNA versus CEA as a predictor of cancer relapse in resected CRC patients and reported poor sensitivity and specificity with CEA [29,32,55].

IDEA established three months of ACT as a standard for most subgroups of early-stage colon cancer [65]. A natural question would be if ctDNA can help determine the optimum duration of adjuvant therapy. No prospective data exist to address this question. However, a post hoc analysis of the PRODIGE-GERCOR IDEA-France trial demonstrated a worse outcome for the ctDNA-positive patients receiving three months of ACT than the group receiving six months of ACT, especially in patients with high-risk stage III colon cancer [66]. Future studies will have to address this critical question.

Adjuvant therapy escalation and de-escalation guided by ctDNA in patients with early-stage CRC is one of the aims of ctDNA technology. To achieve that goal, the ability of the ctDNA test to identify MRD status correctly after the curative-intent surgery would be of prime importance. In other words, a false negative ctDNA test result will seriously undermine the treatment de-escalation strategy. In the study reported by Tie and colleagues with stage II resected colon cancer patients who did not receive ACT, recurrence occurred in 16/164 (9.8%) patients with negative post-operative ctDNA [55]. In the study by Reinert et al., the relapse rate in patients with post-operative ctDNA was 12%, irrespective of ACT administration [29]. The study with the tumor-agnostic REVEAL platform reported recurrence in 12 out of 49 (24.5%) ctDNA-negative patients at the ‘landmark’ time point (defined as one month after the completion of definitive therapy), with a landmark recurrence sensitivity and specificity of 55.6% and 100% [28]. Therefore, the sensitivity of the ctDNA tests must improve for the treatment de-escalation strategy to succeed. One strategy to partly mitigate the sensitivity-related issue is serial testing that can detect more patients with MRD. Indeed, with serial testing, Reinert et al. [29] reported a sensitivity of 88%, and Parikh et al. reported a sensitivity of 91% [28].

## 6. ctDNA-Guided Surveillance

The current surveillance strategy following definitive therapy for patients with early-stage CRC consists of periodic clinical follow-up, serial serum CEA level measurement, CT scans, and colonoscopy [67]. The goal of surveillance is to detect cancer relapse early to enable intervention that might improve outcomes. However, evidence is lacking to prove that the current surveillance protocol improves survival [68,69]. Several observational studies have reported that ctDNA detection in the plasma samples predicts radiologic cancer relapse with a median lead time ranging from 5.5 months to 11.5 months and a positive predictive value close to 100% (Table 2). In the study by Reinert et al., serial ctDNA testing predicted cancer recurrence up to 16.5 months before radiologic imaging (mean, 8.7 months; range, 0.8–16.5 months). Similar findings have been reported in several other prospective studies utilizing a wide variety of ctDNA assay platforms [28,30,32,55,70]. Several studies compared the sensitivity and specificity of ctDNA vs. CEA in predicting cancer relapse and reported that ctDNA outperformed CEA by a significant margin [29,32]. However, the critical question as to whether ctDNA-based early diagnosis of cancer recurrence impacts survival remains unanswered thus far, and will likely be addressed by several ongoing clinical trials (Table 3).

## 7. ctDNA-Guided Clinical Trials

Compelling data published in recent years demonstrated that ctDNA detection in patients with CRC who have completed definitive therapy predicted cancer relapse in virtually all patients [27,28,29,30,32,33,49,55]. The robustness of ctDNA as a biomarker for MRD has injected fresh enthusiasm into the adjuvant therapy research community, as ctDNA technology holds the potential to rescue adjuvant therapy research from decade-long stagnation. Several overarching questions need to be addressed before ctDNA-guided adjuvant therapy can be widely adopted in routine clinical practice: (1) how do we standardize the preanalytical variables to ensure the consistency of test results? (2) Is ctDNA predictive of adjuvant therapy benefit? Although the prognostic value of ctDNA is well established, data are lacking to support that ctDNA is predictive of treatment benefits. (3) Can ctDNA help guide adjuvant therapy escalation/de-escalation post-surgery? Finally, (4) how do we treat the patients who are ctDNA-positive after the completion of adjuvant therapy?

It is well-established that the sensitivity and specificity of ctDNA-based assays are influenced by a variety of preanalytical variables [71]. A detailed discussion of this topic is beyond the scope of the current article, but we would like to inform our readers that well-coordinated efforts are underway to standardize the preanalytical variables by various organizations and expert panel groups, including the US National Cancer Institute appointed task force and the International Liquid Biopsy Standardization Alliance (ILSA), as elaborated in their published white papers [72,73].

Recently presented GALAXY data, as discussed in the previous section, provided preliminary evidence suggesting that ctDNA-guided ACT administration improved survival outcomes in early-stage CRC patients, although the median follow-up was relatively short (11.4 months) [62]. Numerous clinical trials are underway to investigate whether ctDNA-guided ACT administration would improve survival, both in stage III patients (CIRCULATE-US, DYNAMIC-III, VEGA, PEGASUS, ACT-3) and stage II patients (COBRA, DYNAMIC-II, CIRCULATE-PRODIGE 70, NCT04089631, MEDOCC-CrEATE, IMPROVE-IT, etc.), as outlined in Table 3.

Adjuvant therapy de-escalation guided by ctDNA is another primary objective of several trials. Although the GALAXY study demonstrated similar short-term survival in the ctDNA-negative population irrespective of ACT administration [62], robust validation studies are necessary to establish the treatment de-escalation paradigm on a solid footing. This strategy is being investigated in many current trials, including CIRCULATE-US, COBRA, DYNAMIC-III, DYNAMIC-II, etc. (Table 3). Conversely, another set of trials is exploring adjuvant therapy escalation with a triplet regimen (e.g., modified FOLFOXIRI) in ctDNA-positive patients (CIRCULATE-US, DYNAMIC-III), as the efficacy of current adjuvant therapy regimens is modest.

As ctDNA detection post-surgery in CRC patients predicts cancer recurrence in virtually all patients [29,30,74,75,76] without adjuvant therapy, ctDNA is evolving as a potential surrogate efficacy endpoint in many adjuvant trials [77]. ctDNA-guided trial population enrichment will reduce the number needed to treat in adjuvant trials [77], saving valuable resources. Furthermore, ctDNA clearance with ACT as an endpoint could provide an early indication of treatment efficacy in comparison to conventional endpoints such as PFS or OS, saving time in conducting an adjuvant therapy trial. The gain in trial efficiency will also likely translate into expedited approval of new therapies.

The other outstanding question is how do we treat the patients who continue to have detectable ctDNA after the completion of adjuvant therapy, as the detection of ctDNA after ACT invariably predicts cancer recurrence in almost all patients [27,28,31]. These patients should be enrolled in the trials investigating novel therapies targeting MRD before clinically overt cancer relapse (NCT03803553, NCT03436563). These trials will likely pave the way for a new ctDNA-guided treatment paradigm in the CRC treatment continuum.

## 8. Future Perspective

ctDNA-guided adjuvant therapy in early-stage CRC is a novel emerging paradigm that will likely replace the current paradigm, as outlined in Figure 2. The following criteria must be fulfilled before ctDNA can inform adjuvant therapy decisions in routine clinical practice: (1) ctDNA assays must have a high analytical sensitivity and reproducibility to enable ctDNA-guided intervention in the early post-operative period when the likelihood of success is as high as the total micrometastasis burden, and clonal complexity is minimal; (2) ctDNA assays must have a very low rate of false-negative results to enable withholding ACT in ctDNA-negative population (treatment de-escalation); (3) ctDNA assays must have a short turnaround time, preferably two weeks or less; and (4) prospective evidence of survival benefits with ctDNA-guided adjuvant therapy must be provided. To achieve these goals, efforts are underway primarily on two fronts—(a) standardizing preanalytical variables and refining the ctDNA assays to improve sensitivity and reproducibility, including consideration of collecting more plasma, and (b) clinical trials to validate the ctDNA-guided treatment strategies, including treatment escalation and de-escalation, as discussed in the previous section (Table 3).

The sensitivity of ctDNA assays depends on a wide variety of factors, ranging from the preanalytical and analytical variables to tumor biology. Henriksen et al. reported undetectable ctDNA in the preoperative plasma samples in 9% of stage III CRC patients, suggesting that ctDNA shedding characteristics of the tumors also play a role in determining assay sensitivity [27]. The tumor characteristics that can influence the shedding of ctDNA in the bloodstream are numerous, including the location of the tumor, total tumor burden, vascularity, cellular turnover, etc. [78]. For example, higher sensitivity has been reported to be associated with distant recurrence as opposed to locoregional cancer recurrence [31]. To avoid the low shedding-related sensitivity issue, one potential strategy could be excluding the patients with undetectable preoperative ctDNA from MRD assessment. As discussed in the previous section, serial ctDNA testing increases the detection rate. However, it is unknown whether serial testing indeed identifies more patients with MRD or detects early recurrences. Parikh et al. published encouraging prospective data reporting improved sensitivity with the integration of epigenomic signatures in the panel [28], a strategy that has been supported by other studies [79,80]. The integration of emerging techniques such as fragment size analysis, multi-UMIs to minimize PCR errors, background polishing, and the development of a more advanced bioinformatics filter will also likely help improve sensitivity [26].

In early-stage CRC, one of the primary goals of risk stratification after the surgery is to identify patients who do not harbor MRD and should not receive ACT. To that end, integrating multiple prognostic factors may be necessary, an idea proposed by other ctDNA research pioneers [81]. The consensus immunoscore assay is a well-validated prognostic biomarker [65] that might help to refine risk stratification and guide adjuvant therapy decisions if combined with ctDNA. Tarazona and colleagues investigated whether integrating ctDNA, tumor CDX2 expression, consensus molecular subtype (CMS) classification, and inflammation-associated cytokines improves risk-stratification accuracy in a prospective observational study [58]. The study reported that only ctDNA (HR 13.64; *p* = 0.002) and a lack of CDX2 expression (HR 23.12; *p* = 0.001) were independent prognostic factors for DFS in the multivariable model. This concept, however, requires validation by well-designed prospective studies.

As targeted therapies are evolving as powerful tools against many cancers, this paradigm should be extended to MRD targeting. The advances in NGS technology have enabled the identification of actionable genomic alterations in MRD, which are being targeted with novel molecules in various clinical trials (e.g., NCT03803553, NCT04853017, NCT03832569). Another novel approach in this space is targeting MRD with a personalized neoantigen cancer vaccine combined with an immune checkpoint inhibitor (ICI) in proficient mismatch repair (MMR) high-risk stage II or stage III colon cancer patients harboring MRD after the completion of standard ACT (NCT05158621). A similar trial investigates ELI-002, a RAS-targeting vaccine in the MRD setting (Amplify-201) of solid tumors, including CRC harboring RAS-mutated tumors. Another trial investigates the efficacy of an ICI, pembrolizumab, in ctDNA-detectable MMR-deficient solid tumors (NCT03832569). To support the novel drug development targeting MRD, the US FDA has recently published draft guidance for the industry regarding the use of ctDNA for drug development in patients with early-stage solid tumors (https://www.fda.gov/regulatory-information/search-fda-guidance-documents/use-circulating-tumor-deoxyribonucleic-acid-early-stage-solid-tumor-drug-development-draft-guidance, accessed on 20 May 2022). Finally, ctDNA has shown a high prognostic value in all ctDNA assay platforms. However, it is unclear at present if ctDNA is predictive of treatment benefits. Numerous trials are underway to address this vital question (Table 3).

Another critical question would be if ctDNA can provide insight into the biological behavior of cancer. In a prospective observational study with stage III CRC patients, Henriksen et al. reported that the ctDNA growth rate was prognostic for survival (HR = 2.7; 95% CI, 1.1–6.7; *p* = 0.039) [27]. Longitudinal ctDNA analysis in this study revealed a similar OS for the non-relapsing patients and the relapsing patients with a slow ctDNA growth rate (*p* = 0.18). Conversely, patients with a fast ctDNA growth rate had significantly reduced OS (HR = 42; 95% CI, 8–221; *p* < 0.001). Future studies should investigate whether patients with fast ctDNA growth rates should be treated differently.

## 9. Conclusions

An overwhelming amount of data suggest that ctDNA has a high prognostic value and ctDNA-guided risk stratification in patients with resected CRC outperforms the current clinicopathologic criteria-based risk stratification, ushering in numerous clinical trials to investigate ctDNA-guided adjuvant therapy strategies. Prospective clinical trials must establish that ctDNA is predictive of treatment benefits before ctDNA-guided adjuvant therapy is adopted in routine clinical practice. ctDNA technology will also likely provide valuable information regarding the efficacy of adjuvant treatment within a short timeframe, promising the rapid completion of adjuvant trials with novel therapies, enriching our adjuvant therapy armamentarium. The new paradigm of ctDNA-guided treatment of MRD will likely transform CRC treatment, with a profound impact on patient outcomes.

## Figures and Tables

**Figure 1 cancers-14-03078-f001:**
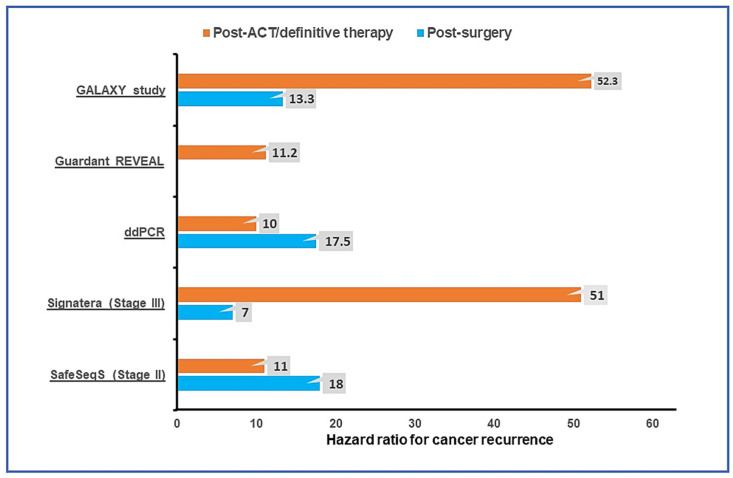
Risk of cancer recurrence reported in prospective observational studies with positive ctDNA in the post-operative and post-adjuvant therapy/definitive therapy settings utilizing tumor-agnostic Guardant REVEAL [28] and tumor-informed ctDNA assays (ddPCR [30], Signatera [27], and SafeSeqS [55], GALAXY study [62]). ACT, adjuvant chemotherapy; ddPCR, droplet digital polymerase chain reaction.

**Figure 2 cancers-14-03078-f002:**
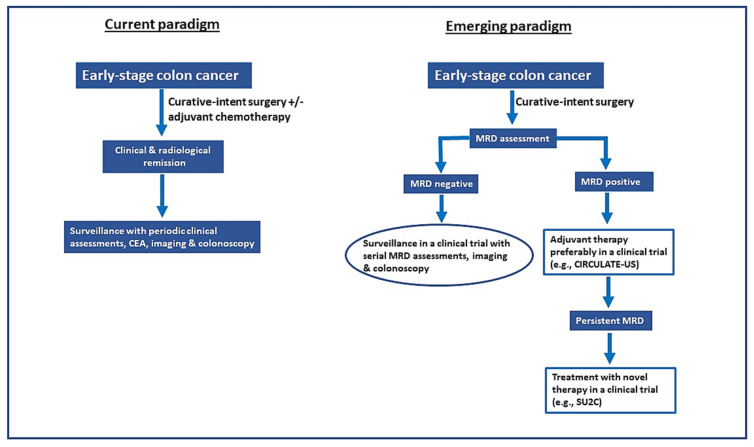
Comparison of current versus the evolving ctDNA-guided adjuvant therapy paradigm in patients with colorectal cancer treated with curative-intent surgery. However, the ctDNA-guided adjuvant therapy strategy needs confirmation through prospective clinical trials before wide adoption in routine clinical practice. CEA, carcinoembryonic antigen; MRD, minimal residual disease.

**Table 2 cancers-14-03078-t002:** Selected studies supporting the value of circulating tumor DNA (ctDNA) for minimal residual disease assessment and surveillance in patients with resected CRC.

Study	Patient Population	*n*	ctDNA Assay	ctDNA Testing Time Points	Major Findings	Comments
Tie et al., 2016 [55]	Stage II CC	230	Safe-SeqS	4–10 weeks post-op and every 3 months for 2 years	Cohort not receiving ACT If ctDNA-+ve post-op: HR for cancer recurrence—18 (95% CI, 7.9 to 40). Cohort receiving ACT If ctDNA-+ve post-ACT: HR for recurrence—11 (95% CI, 1.8 to 68).	ctDNA detection preceded radiologic recurrence by a median of 5.5 months.
Reinert et al., 2019 [29]	Stages I to III CRC	130	Signatera	Preop, post-op day 30, and every 3 months for up to 3 years.	HR for cancer recurrence with positive ctDNA: ₋ Post-op day 30: 7.2 (95% CI, 2.7–19.0) ₋ Shortly after completion of ACT: 17.5 (95% CI, 5.4–56.5). ₋ Serial monitoring post-ACT: 43.5 (95% CI, 9.8–193.5).	Serial ctDNA analyses revealed disease recurrence up to 16.5 months ahead of radiologic imaging (mean, 8.7 months; range, 0.8–16.5 months).
Tie et al., 2019 [32]	Stage III CC	96	Safe-SeqS	4–10 weeks post-op and within 6 weeks of ACT completion	HR for cancer recurrence with positive ctDNA: ₋ Post-op: 7.5 on multivariable analysis (95% CI, 3.5–16.1). ₋ Shortly after ACT: 6.8 (95% CI, 11.0–157.0).	RFS at 3 years in patients who are ctDNA-positive vs. -negative: post-op 47% vs. 76%, post-ACT 30% vs. 77%.
Tarazona et al., 2019 [30]	Stages I to III CC	150	Tumor-informed ddPCR	Preop, 6–8 weeks post-op and every 4 months up to 5 years.	HR for recurrence with positive ctDNA: ₋ Post-op (multivariable adjustment): 11.6 (95% CI, 3.6–36.8). ₋ Post-ACT: 10.02 (95% CI, 9.2–307.3).	ctDNA detection during surveillance preceded radiological recurrence by a median of 11.5 months.
Henriksen et al., 2022 [27]	Stage III CRC	168	Signatera	2–4 weeks post-op and every 3 months thereafter	Detection of ctDNA was a strong recurrence predictor post-o (HR = 7.0; 95% CI, 3.7–13.5) and immediately after ACT (HR = 50.76; 95% CI, 15.4–167).	ctDNA detected recurrence with a median lead-time of 9.8 months compared with radiologic studies.
Parikh et al., 2021 [28]	Stages I–IV CRC	103	Tumor-uninformed assay (REVEAL)	Post-op, post-ACT, and longitudinally in some patients	HR for recurrence when +ve for ctDNA post definitive therapy and with >1 year of follow-up: 11.28.	Integrating epigenomic signatures increased sensitivity by 25–36% versus genomic alterations alone.
Tie et al., 2019 [33]	Locally advanced rectal carcinoma	159	Safe-SeqS	Pretreatment, post CRT, and 4–10 weeks after surgery.	Significantly worse RFS if ctDNA was detectable after CRT (HR, 6.6; *p* < 0.001) or after surgery (HR, 13.0; *p* < 0.001).	The estimated 3-year RFS was 33% for the post-op ctDNA-positive patients and 87% for the post-op ctDNA-negative patients.
McDuff et al., 2021 [59]	Locally advanced rectal carcinoma	29	ddPCR	Baseline, preop, and post-op	At a median follow-up of 20 months, patients with detectable post-op ctDNA experienced poorer RFS (HR, 11.56; *p* = 0.007).	All patients (4 of 4) with detectable post-op ctDNA recurred, whereas only 2 of 15 patients with undetectable ctDNA recurred (negative predictive value = 87%).
Tie et al., 2021 [60]	CRC with liver metastasis	54	Safe-SeqS	Preop and post-op samples, serial samples during pre- or post-op chemotherapy, and follow-up	Detectable post-op ctDNA predicted a significantly lower RFS (HR, 6.3; 95% CI, 2.58 to 15.2; *p* < 0.001) and OS (HR, 4.2; 95% CI, 1.5 to 11.8; *p* < 0.001)	End-of-treatment (surgery +/− ACT) ctDNA detection was associated with a 5-year RFS of 0% compared to 75.6% for patients with an undetectable end-of-treatment ctDNA (HR, 14.9; 95% CI, 4.94 to 44.7; *p* < 0.001).
Loupakis et al., 2021 [61]	CRC undergoing liver resection	112	Signatera	Post-op, at the time of radiologic relapse or last follow-up.	ctDNA-positive status was also associated with an inferior overall survival: HR: 16.0; 95% CI, 3.9 to 68.0; *p* < 0.001.	ctDNA was detectable in the post-op sample in 54.4% (61 of 112) of patients, of which 96.7% (59 of 61) progressed at the time of data cutoff (HR: 5.8; 95% CI, 3.5 to 9.7; *p* < 0.001).
Kotaka et al., 2022 (Galaxy study) [62]	Stages I–IV CRC patients	1365	Signatera	Before surgery, 1-month post-op and every 3 months thereafter for 2 years	Six-month DFS rate was significantly higher in patients whose ctDNA was converted with ACT compared to patients who remained positive after ACT with an HR of 52.3 (95% CI: 7.2–380.5; *p* < 0.001), after a median follow-up of 6.6 months.	Cumulative incidence of ctDNA clearance was significantly higher with ACT vs. non-ACT (67% vs. 7% by 24 weeks; cumulative HR = 17.1; 95% CI: 6.7–43.4, *p* < 0.001).
Tie et al., 2022 (DYNAMIC) [63]	Stage II CC	455	SafeSeqS	4 and 7 weeks post-surgery	Adjuvant therapy guided by ctDNA resulted in chemotherapy administration in lower proportion of patients without any detriment to 2-year RFS.	DYNAMIC is the first reported prospective randomized study supporting the ctDNA-guided adjuvant therapy approach in stage II colon cancer.

Abbreviations: CC, colon cancer; *n*, number of patients; Preop, preoperative; Post-op, postoperative; Post-ACT, after adjuvant chemotherapy; ctDNA, circulating tumor DNA; Safe-SeqS, safe sequencing system; HR, hazard ratio; CI, confidence interval; MRD, minimal residual disease; CRC, colorectal cancer; RFS, relapse-free survival; ddPCR, digital droplet polymerase chain reaction; NGS, next-generation sequencing; CRT, chemoradiotherapy; DFS, disease-free survival.

**Table 3 cancers-14-03078-t003:** Selected clinical trials investigating the ctDNA-guided treatment strategies in patients with colorectal cancer *.

Study Identifier	Study Phase	Study Population	*n*	ctDNA Assay	Study Description	Primary Endpoint
NCT04068103 (COBRA)	II/III	Stage II CC without high-risk features	1408	LUNAR-1 (Guardant Health)	Arm A: active surveillance. Arm B: ctDNA directed therapy (ctDNA-positive → FOLFOX/CAPOX for 6 months, ctDNA-negative → active surveillance)	Clearance of ctDNA with ACT (phase II) and RFS (phase III)
NCT05174169 (CIRCULATE-US)	II/III	Stage II and III CC	1912	Signatera	Cohort A: Arm 1—ctDNA-negative treated with FOLFOX 3–6 months/CAPOX 3 months. Arm 2—ctDNA-negative undergoing serial ctDNA monitoring and no treatment. Cohort B: Arm 3—ctDNA-+ve treated with FOLFOX/CAPOX for 6 months. Arm 4—ctDNA-+ve treated with mFOLFIRINOX	TTPos (time from randomization until ctDNA-positive event), DFS
NCT04120701 (CIRCULATE-PRODIGE 70)	III	Resected Stage II CC	1980	ddPCR	ctDNA-positive → randomized (2:1) to receive ACT or no ACT. ctDNA-negative → surveillance.	3-year DFS in ctDNA-positive patients.
ACTRN12615000381583 (DYNAMIC-II)	III	Stage II CC	450	Safe-SeqS	Arm A: positive for ctDNA → ACT, negative for ctDNA → surveillance. Arm B: treated at the discretion of the clinicians.	RFS
ACTRN12617001566325 (DYNAMIC-III)	II/III	Stage III CC	1000	Safe-SeqS	Arm A: standard of care. Arm B: ctDNA-informed (ctDNA-negative → therapy de-escalation; ctDNA-positive → therapy escalation)	RFS
GALAXY (UMIN000039205)	Prospective observational	Stages II∓IV CRC	2500	Signatera	Serial ctDNA monitoring after surgery. If ctDNA-negative--> VEGA trial (therapy de-escalation). If ctDNA-+ve → ALTAIR trial.	DFS
VEGA (jRCT1031200006)	III	High-risk stage II or low-risk stage III CC	1240	Signatera	Designed to compare adjuvant CAPOX for 3 months vs. observation for GALAXY patients with negative ctDNA at week 4 after surgery.	DFS
NCT04089631 (CIRCULATE AIO-KRK-0217)	III	Stage II CC	4812	NGS	Patients positive for ctDNA post-resection are randomized to observation vs. capecitabine or CAPOX (investigator’s choice) × 6 months	DFS
MEDOCC-CrEATE (NL6281/NTR6455)	III	Stage II CC	1320	PGDx elio™	Standard of care surveillance vs. ctDNA-guided ACT (ctDNA-positive: 6 months of CAPOX; ctDNA-negative: surveillance)	The proportion of patients receiving ACT after surgery if ctDNA-positive.
NCT03748680 (IMPROVE-IT)	II	Stage I and II CRC	64	ddPCR, NGS	ctDNA-positive patients randomized to observation vs.FOLFOX or CAPOX for 6 months.	DFS
NCT04264702 (BESPOKE)	Prospective Observational	Stage II and III CRC	1000	Signatera	Serial ctDNA testing following surgery and ACT vs. observation at the discretion of the treating clinician. Control arm-matched stage II and III patients with a minimum 2 years of follow-up data.	1. Impact of ctDNA on adjuvant treatment decisions. 2. Rate of ctDNA detected recurrence while asymptomatic.
NCT04259944 (PEGASUS)	II	Resected MSS stage III and high-risk stage II (T4N0) CC	140	LUNAR-1 (Guardant Health)	ctDNA-guided ACT. (i) ctDNA-positive → CAPOX for 3 months; (ii) ctDNA-negative → capecitabine for 6 months but will be retested after 1 cycle, and if found ctDNA-positive, will be switched to CAPOX.	The number of patients negative for ctDNA post-op and post ACT later turning ctDNA-positive or developing radiographic relapse.
NCT04084249 (IMPROVE-IT2)	III	Stage III or high-risk stage II CRC	254	ddPCR, NGS	Patients were randomized to ctDNA-guided post-operative surveillance or standard-of-care CT-scan surveillance.	Fraction of patients with relapse receiving curative resection or local treatment
NCT03803553 (ACT-3)	III	Stage III CC	500	LUNAR-1 (Guardant Health)	ctDNA-enriched second-line adjuvant therapy: patients are distributed post-ACT as follows- 1. ctDNA-negative: active surveillance; 2. ctDNA-positive: (a) MSS patients- 6 months of FOLFIRI vs. active surveillance, (b) MSI high-6 months of nivolumab, (c) BRAF mutant and MSS-6 months of BRAF directed therapy.	DFS, ctDNA clearance rate
NCT04457297 (ALTAIR)	III	Stage II∓IV CRC	240	Signatera	TAS-102 for 6 months vs. placebo for patients positive for ctDNA following completion of standard ACT and no evidence of relapse radiologically in the GALAXY study	DFS
NCT03436563	Ib/II	Stage IV CRC	74	Signatera	Patients with metastatic CRC positive for ctDNA following resection of all metastases will receive M7824 (anti-PDL1/TGFbetaRII fusion protein) for 6 doses	ctDNA clearance rate
NCT04589468	Ia/b	CRC, breast, and prostate cancer stage I–III	70	Signatera	Patients with primary breast, prostate, or colorectal cancer and detectable ctDNA (*n* = 50) post definitive treatment will perform one of five escalating dose levels of exercise.	RP2D of exercise
NCT04853017 (Amplify-201)	I/II	Solid tumors, including CRC with RAS mutation	18	Signatera	Patients with tumors harboring RAS mutation and minimal residual disease with detection of ctDNA receive different dose levels of ELI-002, a RAS targeting vaccine.	1. MTD of ELI-002 and the RP2D 2. Safety

Abbreviations: ctDNA, circulating tumor DNA; CC, colon cancer; ACT, adjuvant chemotherapy; mFOLFOX6, 5-Fluorouracil, leucovorin, and oxaliplatin; ddPCR, droplet digital polymerase chain reaction; NGS, next-generation sequencing; RFS, relapse-free survival; CAPOX, capecitabine and oxaliplatin; DFS, disease-free survival; MSS, microsatellite stable; Safe-SeqS, safe sequencing system; CRC, colorectal cancer; MTD, maximum tolerated dose; RP2D, recommended phase 2 dose. * Clinicaltrials.gov accessed between 20 April 2022 and 20 May 2022.

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
