# Peer review of "Finding Waldo: The Evolving Paradigm of Circulating Tumor DNA (ctDNA)—Guided Minimal Residual Disease (MRD) Assessment in Colorectal Cancer (CRC)"

_cancers, 2022, doi:10.3390/cancers14133078_

Round 1

Reviewer 1 Report

I enjoyed reading your report that mentioned ct DNA in CRC patients.  As I always consider the ways how makes recurrence of CRC patients and how reduces adverse events of adjuvant chemotherapy, your report gave me much impact regarding of the possibility to answer my clinical questions. And  I would like to know some more questions about adjuvant therapy; first, ctDNA of CRC patients who have undergone resection of their tumor can predict cancer recurrence perfectly other than tumor marker (CEA)?  Second, the length of adjuvant therapy should be determined with ct DNA? Adjuvant therapy should be continued until MRD negative? Third, adjuvant chemotherapy for high risk Stage II elderly patients (over 80s) should be performed?  I expect that next reports will answer my questions.

Author Response

Please see the file uploaded. Thank you.

Reviewer 2 Report

Undetectable residual cancer cells by standard tests and imaging studies are responsible for cancer recurrence after surgical therapy in localized colorectal cancer.

Circulating tumor DNA indicates the presence of residual cancer cells. The current article discusses how ctDNA technology can help to recommend adjuvant therapy with serial assessments to determine if we need to continue or not with treatment. However, it would be appreciated any insight about which therapy to be recommended. May the ctDNA analysis provide clues for any specific adjuvant therapy? Must we treat all patients positive for ctDNA in the same way?

Author Response

(The authors gave the same response as above.)
